# Protocol for a scoping review on information and communication technologies (ICTs) in community practice

**Ofir Sivan** [ID]°*, **Ali Pearson**°, **Stephanie Begun** [ID]°

Factor-Inwentash Faculty of Social Work, University of Toronto, Toronto, Ontario, Canada

☯ These authors contributed equally to this work.
* ofir.sivan@mail.utoronto.ca

## Abstract

### Introduction

There is a growing literature connecting information and communication technologies (ICTs) to community-based practices including community organizing, community development, and participatory planning. Over a decade ago, the American Academy of Social Work and Social Welfare launched the Grand Challenge to Harness Technology for Social Good. Despite its potential to answer this challenge, community practice is absent from all systematic reviews on social work and technology. The proposed scoping review will address this gap by identifying, organizing, summarizing, analyzing, and synthesizing the existing literature connecting community practice and ICTs in social work and related disciplines.

### Methods

This protocol was developed using Arksey and O'Malley's [2005] framework and the Preferred Reporting Items for Systematic Review and Meta-Analysis Protocols [PRISMA-P]. Search will be conducted in the following databases: Sociological Abstracts [ProQuest interface, 1952-], [Social Service Abstracts ProQuest interface, 1977-], Applied Social Service Index Abstracts [ProQuest interface, 1987-], Social Work Abstracts [Ovid interface, 1968-], Communication Abstracts [EBSCO interface, 1915-], Social Science Citation Index [Web of Science interface, 1900-], and Emerging Sources Citation Index [Web of Science interface, 2005-]. Published records eligible to include in this scoping review include conceptual and empirical peer-reviewed articles. Selection criteria, and methods of data extraction, management, and analysis are detailed.

**Data availability statement:** All relevant data are within the paper and its Supporting information files.

**Funding:** The author(s) received no specific funding for this work.

**Competing interests:** The authors have declared that no competing interests exist.

## Discussion

The proposed scoping review will explore the engagement of community practitioners with ICTs. This review will also identify ICT-facilitated community-based practices that the social work profession can reclaim in a bid to address the Grand Challenge to Harnessing Technology for Social Good. Limitations, implications for future research, and methods of disseminating the results are discussed.

## Introduction

Information and communication technologies (ICTs) refer to a wide range of online and offline technologies used to store, process, and disseminate information and/or support communication [1,2]. These technologies include, but are not limited to, computers, telephones, mobile devices, social media, algorithms, big data analytics, artificial intelligence, and geospatial technologies [3]. Over the past three decades, ICTs have become an integral part of social work research [4], education [5], and practice [6–8], shaping its epistemological assumptions [9,10] and professional ethics [10,11]. To date, several scoping and systematic reviews have been conducted to map the relationships between social work and technology [12]; evaluate the empirical evidence produced in technology-mediated intervention studies in social work [13]; and assess technology-based interventions in social work practice in general [14], specifically in the area of mental health social work [15]. Additional scoping and systematic reviews mapped and assessed the effectiveness of ICTs (such as web-based instruction methods or virtual-reality simulation) in social work education [16–18]. Despite growing research on the relationship between ICTs and community-based practices in both social work [19–21] and related social science disciplines [22–25], no scoping or systematic reviews have mapped, analyzed, and synthesized this literature. Scoping and systematic reviews on community-based practices are rare, and those that do exist contain little to no reference to ICTs [26–29].

Social work scholars have repeatedly raised community practice as a way to counter populism and the erosion of liberal democracy institutions [30–35], build democracy through citizen participation [36,37], and advocate for a more equitable redistribution of power, services, and resources [22,38,39]. The significant decline in North American social work's engagement with community practice has been associated with the dispersion of community practice research across various social sciences and with a lack of systematic investigation from a social work perspective into how ICTs are used to facilitate and implement community practices [38]. To address this gap, we propose a scoping review that will locate, organize, analyze, and synthesize the extensive interdisciplinary literature connecting community practice and ICTs.

### Community practice in social work

This scoping review will adapt the North American social work's definitions and classifications of community practice. Gutiérrez and Gant [2018] define community practice as any method of intervention that: (a) engages communities or community

organizations; (b) aims to motivate change in the community or larger social systems; and (c) aligns and advances social work ethics and values such as "social justice, the dignity and worth of individuals, the importance of human relationships, integrity, and competence" [38] (p. 618). While this definition provides the necessary conditions for community practice, scoping reviews require more exclusive and exhaustive definitions for developing search and screening strategies [40,41]. For this goal, we build on various typologies of community practice offered by social work scholars [36–39,42–47] to define three core strategies of community practice: community organizing, community development, and participatory planning. Each of these strategies can be implemented using various intervention methods. That said, it is important to remember that the division between the three strategies is somewhat artificial since community practitioners usually combine more than one strategy to achieve their goals [36,48,49].

The concept of community in community practice is also diverse, and may refer to: (a) functional communities that are temporarily organized for a specific mission or purpose; (b) communities of interest that are organized around practicing faith, sports, or education; (c) local, place-based communities both in rural and urban settings; (d) virtual communities such as those created on social media and the internet; (e) identity-based communities such as the LGBTQ+ community, women's groups, and immigrant communities; and (f) communities built around lasting social problems such as unequal education, unaffordable housing, and inaccessible health service provision [36,37,43,45,46]. The community strategy used is closely related to the concept of community itself. For example, community organizing is often connected with functional communities, which can exist in either physical locations or virtual spaces. On the other hand, community development is more effectively carried out in place-based communities.

**Community organizing.** Community organizing is an umbrella term for various community-based practices sharing the assumption that conflict, disruption, and pressure are a legitimate means to advance community goals [44]. The guiding principle of this strategy is that organizing, of both people and organizations, should be done 'bottom-up', allowing community members and organizations to take various roles and responsibilities [46,49,50]. In community organizing the authority to make decisions and set goals is determined by the community members, leaders, and activists, rather than by professionals or administrations [48,51,52]. Community organizing is action-oriented and usually includes one or more of the following methods: grassroots organizing [53]; collective mobilization [36]; lobbying and advocacy [48]; civil disobedience, boycotts, strikes, and demonstrations [38,43]; group- or organization-based coalition building [22,37]; educational campaigns [37,45]; and Freirean critical pedagogy, conscientization, and teach-ins [54,55].

Community organizing can vary significantly in its goals. Some initiatives may focus more on symbolic aspects such as getting recognition or building community's social capital [46], while others emphasize achieving material objectives, such as ensuring equal access to social services and resources [39]. community organizing's efforts may differ in their level of reliance on sponsoring organizations or professionals, such as social workers, policymakers, and lawyers [38], as well as in the extent to which service provision is integrated into the organizing activities [37]. It is important, however, to distinguish community organizing from both social activism and mass mobilization within social movements. Unlike activism, community organizing aims at bringing community members together for a planned, collaborative, and democratic decision-making process [48]; and while community organizing can sometimes emerge from the institutionalization of social movements, it typically results from a long-term, systematic, and planned community building process rather than a spontaneous reaction to specific events [43].

**Community development.** Community development is a strategy that focuses on building community capacity and leadership, emphasizing the importance of consensus-building through community participation [36,44,48]. This strategy assumes that communities can solve their problems when provided with the right skills and sufficient access to resources and opportunities [36,45]. The main goal of community development is sustainable social reproduction [49] that can address social, economic, and environmental disparities through service provision [37,56]. Heavily relying on professional expertise and leadership, community development practitioners incorporate practices such as general and vocational adult education, community sensitization, and awareness-raising regarding local problems, as well as participatory service

development, delivery, and evaluation [37,43,48]. Since its goal is to connect social needs with available resources to support sustainable economic development [38], community development is generally open to the involvement of external stakeholders, including local developers, government foundations, charities, banks, and international organizations [37].

While both community development and community organizing share a critical analysis, placing the origins of community problems in unequal access to resources and opportunities, each of these strategies envisions the solution differently [45]. Community organizing typically focuses on mobilizing the community members against specific policies, institutions, or individuals perceived as responsible for the injustices they experience [50]. In contrast, community development aims to equip communities with skills and resources, enabling them to solve their own problems while emphasizing consensus between communities and social institutions. [36]. Some scholars, however, suggest that community organizing methods can serve as a means to secure resources for community development initiatives [48,49].

**Participatory planning.** Participatory planning, also known as citizen participation [43] or inclusive program development [37], is a process that actively engages communities in identifying a shared vision for change, conceptualizing the necessary conditions for that change to happen, and outlining the steps required to achieve it [36,37,43,45]. Unlike community organizing and community development, social planning—even when participatory— follows a 'top-down' strategy [36]. As such, it perceives community as something that exists rather than something that needs to be created. Like traditional forms of social planning, participatory planning emphasizes the importance of data, empirical evidence, and experts [36]. However, participatory planning also incorporates community members, citizens, residents, and community organizations in these 'top down', data-driven social planning efforts, usually through multi-sector collaborative governance of social programs and public policy consultations [36,37,43].

## The grand challenge context: Harness technology for social good

Over a decade ago, the American Academy of Social Work and Social Welfare (AASWSW) brought together leading experts and scholars in social work to launch the Grand Challenges for Social Work (GCSW) framework [57–59]. The purpose of GCSW was to encourage interdisciplinary research focused on what were seen as the most pressing issues facing social work practice, theory, and education [58]. The challenge to *Harness Technology for Social Good* was conceptualized under the category of creating a 'stronger social fabric', along with 'end homelessness' and 'eradicate social isolation' [59]. Two working papers were published in 2015, outlining the barriers for- and potential of ICTs [60] and big data [61] to address wicked social problems and advance social work practice, research, and education. Three book chapters evaluating the progress of social work in addressing this Grand Challenge indicate the profession's primarily focus on harnessing technology in micro-level (i.e., clinical interventions with individuals and families) and the macro-level (i.e., planning and policymaking) practices [59,62,63]. A decade after the launch of GCSW provides a valuable opportunity to examine the potential of social work to effectively harness technology for social good, by exploring the relationships between ICTs and community practice.

## Study objectives

The primary goal of the proposed scoping review is to identify, describe, analyze, and synthesize the extensive literature connecting ICTs with community practice. Specifically, this scoping review will: (a) map and categorize this literature (e.g., discipline, theoretical framework, methodology, geography); (b) describe the relationships between community practice and ICTs; (c) indicate gaps in the literature and questions for future research/focused systematic reviews; and (d) identify ICT-facilitated community practices that can be reclaimed by the social work profession.

## Methods

We will follow Arksey and O'Malley's [41] methodology for scoping reviews which includes five stages: (a) identifying research questions (b) locating the relevant literature, (c) selecting the literature, (d) charting the data and (e) collating,

summarizing, and reporting the results. This protocol was developed by employing the Preferred Reporting Items for Systematic Review and Meta-Analysis Protocols [PRISMA-P] checklist [64] (S2 Appendix). This scoping review will be conducted by a doctoral student (OS), a doctoral candidate (AP), and an Associate Professor (SB)—all from the Factor Inwentash Faculty of Social Work at the University of Toronto. The research protocol and search strategy methods were developed by a doctoral student (OS) in consultation with the other team members (SB, AP) and social work librarian at the University of Toronto (YL; see Acknowledgements). The research team approved the protocol and its stated objectives.

## Formulating research questions

Following Arksey and O'Malley's [41] recommendation, we aim to cover a broad scope of the literature connecting community practice and ICTs as defined in the Introduction. Accordingly, our overarching research question is: what is known, based on the published interdisciplinary and peer-reviewed literature, about the relationship between community practice and ICTs? Specifically, we ask: (a) How are the relationships between community practice and ICTs described in this literature? (b) How are ICTs used in community practice, and what role does community practice play in addressing the social impacts of ICTs? (c) What definitions of community and community practice can be identified from this literature? (d) What gaps and areas for future research exist in this literature? (e) What ICT-facilitated community-based practices can be reclaimed by the social work profession?

## Identifying relevant literature

The initial search strategy was developed by the first author (OS) and has been iteratively refined via consultations with the social work librarian (YL) and the research team (AP, SB). We used an initial bibliographical list of 30 journal peer-reviewed articles addressing the relationships between community practice and ICTs that have been published since 2015 to determine the final databases for search and test our search strategy. We scanned these 30 articles, alongside indices of community practice, digital sociology, digital social work handbooks [7,37,53,65–68] and scoping reviews on technologies and social work [12–15,17,18] to develop our initial search strategy. We optimized our search strategy using both controlled vocabulary (ProQuest Sociological Thesaurus search) and natural language (keywords) to capture maximum variations in our terminology. Additionally, our search strategy included both Boolean and proximity operators (e.g., NEAR/2), truncation, and wildcards to cover common phrases such as "community-based organizing" (see S1 Appendix). The final search strategy was developed and tested in Sociological Abstracts (ProQuest interface, 1952-) using both keyword and the thesaurus search (see the complete search string attached in Appendix S1). Search strategy will be translated to perform the final search using the following databases: Sociological Abstracts (ProQuest interface, 1952-), Social Service Abstracts (ProQuest interface, 1977-), Applied Social Service Index Abstracts (ProQuest interface, 1987-), Social Work Abstracts (Ovid interface, 1968-), Communication Abstracts (EBSCO interface, 1915-), Social Science Citation Index (Web of Science interface, 1900-), and Emerging Sources Citation Index (Web of Science interface, 2005-). These databases will enable us to comprehensively identify the literature published in social work and related fields, including education, psychology, sociology, urban studies, development studies, and communication. To ensure thorough coverage of the literature, we will also perform a supplementary search in the reference lists of the articles screened for extraction, as well as these articles' journal websites. Search strings, dates, and result numbers for each database will be recorded in a table to help the research team track future modifications. Searches in all databases will be restricted to publications in English between 2015 and 2025. We chose this timeframe considering the Grand Challenge of *Harness Technology for Social Good* was launched in 2015. Recently, Hitchcock et al. (2024) published a scoping review of the scholarship published by social work scholars and/or in social work journals to assess the advancement of the profession in meeting this Grand Challenge [12]. Their review, however, does not specifically cover the relationship between community practice and technology, is limited to social work literature, and included publications only between 2015–2019. Recognizing the multi- and

interdisciplinarity of community practice, our review aims to cover the literature published during the decade passed since the launch of the Grand Challenges.

## Screening eligible literature

The literature identified using the search strategy outlined above and in S1 Appendix will be downloaded from databases as RIS files (including titles and abstracts) and uploaded to Covidence, a web-based software used for conducting systematic and scoping reviews, for deduplication and screening. At the first stage, title and abstract screening will be conducted independently by the first two authors (OS, AP) using the following eligibility criteria: (a) written in English; (b) published between 2015–2025; (c) published in peer-reviewed journals; (d) include reference to both ICTs and community practice as conceptualized in the Introduction; (e) can be empirical, conceptual, and theoretical articles; (f) include conceptual/theoretical articles without geographical restriction; and (h) restricted to empirical studies with data collected/ generated in the United States or Canada. This geographical restriction is based on three considerations: historical, professional, and theoretical. Historically, community practice in the Anglo-American social work (especially in the United States, United Kingdom, and Canada) emerged from the settlement houses at the beginning of the twentieth century and has been shaped by the neoliberal structuring of the welfare state [45,69]. In contrast to the UK, Canada and the United States share a more decentralized and flexible model of welfare provision, emphasizing the roles of civil society, community initiatives, and nonprofits rather than state control [70,71]. Professionally, both the United States and Canada implement similar accreditation and state- or province-controlled regulation, while in the UK the profession is regulated by a single statutory body [72]. These historical and professional contexts have led to different theoretical emphases in community social work on both sides of the Atlantic. In the UK, a state-sponsored model of community development emerged [73,74], while in the United States and Canada there were greater opportunities for grassroots organizing, local entrepreneurship, community activism, and social movements [50,53,69]. Upon completing the title and abstract screening, the first two authors (OS, AP) will independently complete a full text review of the selected articles. Conflicts and discrepancies during the full text review will be resolved by the third author (SB).

## Charting the data

We will systematically categorize and organize data from the articles meeting the eligibility criteria using five extraction tables, each featuring a Covidence article identifier in the leftmost column to enable tracking and support data analysis and synthesis. While the extraction will be performed by the first author (OS), research team members will pilot this extraction framework with the bibliographical list of 30 articles used to select data bases and test the search strategy. Extraction categories will be modified if necessary.

   The first table will include the following bibliographical details: (a) authors' names; (b) first author disciplinary/professional affiliation; (c) first author country and state; (d) article title; (e) year of publication; (f) journal name; (g) journal's disciplinary categorization [in Web of Science]; (h) whether the article is conceptual or empirical; (i) methodological design for empirical studies [i.e., qualitative, quantitative, mixed methods]; (j) community practice strategy [i.e., community organizing, community development, participatory planning]; (k) type of ICT [e.g., artificial intelligence, social media]; and (l) Digital Object Identifier [DOI]/Uniform Resource Locator [URL].

   The second table will organize and categorize the following additional data from conceptual articles: (a) theoretical framework; (b) conceptualization of community practice; (c) conceptualization of community; (d) authors' approach regarding the use of ICTs in community practice; (e) paper's main arguments/concepts.

   The third table will organize and categorize the following additional data from empirical studies: (a) the specific operationalization/method of community practice [e.g., adult education, campaigns, resident consultations, protests] (b) the professional identity of practitioners or the absence thereof; (c) the type of sponsoring organization or the absence thereof; (d) the main sociodemographic characteristics of the community; (e) the issue or need addressed by the practice; (f) the

 

purpose/rationale of using ICTs in community practice; (g) the ways ICTs are used in community practice; (h) the way the relationship between ICTs and community practice is described; (i) the way ICTs shape different aspects of community practice [e.g., community participation, practitioner motivation]; and (j) country and state/province of data collection [i.e., Canada, the United States].

The fourth table will organize and categorize the following additional data, specifically relating to qualitative papers: (a) research questions; (b) methods of data collection and analysis/design [e.g., ethnography, grounded theory]; (c) guiding theoretical frameworks; (d) sample size; (e) sampling methods; and (f) main findings/themes/trends.

The fifth table will organize and categorize the following additional data, specifically relating to quantitative and mixed methods papers: (a) sample size; (b) sampling methods; (c) research questions/hypotheses (d) study design [e.g., longitudinal, cross-sectional, RCT]; (e) statistical analysis methods; (f) type/level of evidence; (g) results/findings; and (h) conclusions.

Finally, an additional table/document will be dedicated to document our reflections, insights, and key takeaways during the extraction process.

## Data analysis and synthesis

We will perform both quantitative and qualitative analyses. For the quantitative analysis, we will assign numeric values for the extracted categorical data (e.g., methodology, professional affiliation) or continuous data (e.g., year of publication, sample size). Using statistical software, we will describe these variables using univariate statistics and explore the relationships between these variables using bivariate statistical tests (e.g., Chi-square, Pearson correlation, t-tests). For example, we will explore the relationships between the type of ICT used (e.g., social media, big data analytics) and the community practice strategy (e.g., community development, community organizing) using Chi-square test or the correlation between year and number of publications using Pearson correlation. We will employ thematic analysis to summarize and organize the qualitative data extracted from the articles [75]. To begin, we will familiarize ourselves with the data by reading and re-reading it. Then, we will develop a structured codebook to guide our analysis. The first author (OS) will create the initial version of the codebook, which will then be revised and amended by the second and third authors (AP, SB). We will utilize both deductive (theory-driven) and inductive (data-driven) coding methods in this process. Finally, we will organize the codes into a framework and identify themes, sub-themes, and their interrelationships. Additionally, we will synthesize qualitative and quantitative findings [41] to better understand the relationships between community practice and ICTs, identify definitions of community and community practice that emerge from this literature, evaluate the knowledge produced on these relationships since 2015, identify gaps and questions to guide future research, and specify what ICT-facilitated community practice social work can reclaim.

## Ethics

Information for this review will be sourced from publicly available resources. Therefore, ethics approval is not necessary.

## Discussion

This protocol describes a proposal for a scoping review on the relationships between community practice and information and communication technologies (ICTs) in social work and related disciplines/professions. We contextualized the need for this scoping review by showing that, despite the growing interdisciplinary literature connecting community practice and ICTs, community practice is absent from systematic reviews on social work and technologies. Due to the various (and often ambiguous) definitions for community practice, we used community social work definitions and typologies to articulate the conditions under which an intervention is deemed community practice. Following Arksey and O'Malley's [41] methodology for scoping studies and the PRISMA-P checklist guidelines [64], we defined the research questions for this review, developed a thorough search strategy, established selection criteria for studies, outlined extraction categories, and determined methods for both quantitative and qualitative data analysis.

## Implications for theory and practice

The proposed scoping review aims to systematically describe, analyze, and evaluate the growing body of literature connecting community practice and ICTs. It also seeks to address the possibilities, challenges, and roles of community practice in increasingly technologized environments, especially in a political context of growing polarization, populism, violence, and injustice. This review will contribute to community social work theory, education, and practice by examining the field's relationships with technology over the past decade. This scoping review will contribute to the development of innovative and reflexive approaches to planning, implementing, and evaluating ICT-facilitated community-based practices. Our findings will also indicate existing gaps in the interdisciplinary research on the use of ICTs in community practice. Conducted by social work researchers with extensive backgrounds in community practice, this review will be a significant first step towards embracing social work's professional and disciplinary responsibility for community practice. Finally, we contextualized this review within the Grand Challenge framework introduced over a decade ago [58] to emphasize the urgent need to understand how technology can be used to advance social work goals and values [12,59–63]. As social work scholars and community practitioners, we recognize the potential of both community practice and scientific research to provide sustainable solutions and innovative approaches to complex social problems. Therefore, we are committed to the collective effort in our discipline to explore how ICT-facilitated community practice can be harnessed for social good.

## Limitations

This scoping review has three main limitations related to the article selection criteria (i.e., screening). First, the empirical studies covered in this review will be limited to those that collected or generated data in the United States and Canada. These two countries share similar histories of community practice, terminology, and typologies dating back to the late nineteenth century. They also have comparable social work practices and regulatory mechanisms and have similarly responded to the neoliberal restructuring of the welfare state. Although our geographic restrictions are anchored in historical, professional, and theoretical considerations, they were also driven by feasibility. As a result, this scoping review excludes important empirical studies that could contribute to our understanding of the relationship between community engagement and ICTs. Future reviews may benefit from including and comparing empirical studies conducted in other Anglo-American contexts, such as the UK, Australia, and New Zealand.

Second, although social work and related disciplines are increasingly publishing in English-language international journals, this review's exclusion of articles published in other languages may omit conceptual papers in local scholarly languages and empirical studies by francophone scholars in Canada.

Finally, this review does not include book chapters, theses/dissertations, and gray literature, such as reports from local and international nonprofit organizations (NPOs) or non-governmental organizations (NGOs). Future reviews should prioritize collecting this literature, as it has the potential to shed light on additional aspects of ICT use in community practice.

## Reporting and disseminating the results

A final report will be created to present both qualitative and quantitative findings. The report will address all objectives and research questions outlined in this protocol, focusing on both the description, analysis, and evaluation of the current body of knowledge regarding community practices and ICTs. The report will identify trends and gaps in this broad interdisciplinary literature, propose new research questions, and highlight areas that require further exploration through more focused systematic reviews. This report's findings will be disseminated through peer-reviewed journals and national and international conferences. Any amendments to this protocol will be documented in the final publication of the review.

## Supporting information

**S1 Appendix. Search strategy for Sociological Abstracts [ProQuest].**
(DOCX)

**S2 Appendix. PRISMA-P (Preferred Reporting Items for Systematic review and Meta Analysis Protocols) 2015 checklist: Recommended items to address in a systematic review protocol\*.**
(DOCX)

## Acknowledgments

The research team members would like to express their gratitude to Yoonhee Lee, the social work librarian at Robarts Library's user service, University of Toronto for the assistance in developing and refining the search strategy.

## Author contributions

**Conceptualization:** Ofir Sivan, Ali Pearson, Stephanie Begun.

**Data curation:** Ofir Sivan, Ali Pearson.

**Formal analysis:** Ofir Sivan.

**Methodology:** Ofir Sivan, Ali Pearson, Stephanie Begun.

**Project administration:** Ofir Sivan.

**Supervision:** Stephanie Begun.

**Writing – original draft:** Ofir Sivan.

**Writing – review & editing:** Ofir Sivan, Ali Pearson, Stephanie Begun.

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
