## [Decision Letter · Decision Letter 0]

11 Mar 2026

Dear Dr. Sivan,

Thank you for submitting your manuscript to PLOS ONE. After careful consideration, we feel that it has merit but does not fully meet PLOS ONE’s publication criteria as it currently stands. Therefore, we invite you to submit a revised version of the manuscript that addresses the points raised during the review process.

We look forward to receiving your revised manuscript.

Kind regards,

Wei Lang, Ph.D.

Academic Editor

PLOS One

Journal Requirements:

2 If the reviewer comments include a recommendation to cite specific previously published works, please review and evaluate these publications to determine whether they are relevant and should be cited. There is no requirement to cite these works unless the editor has indicated otherwise.

Reviewer's Responses to Questions

**Comments to the Author**

1. Does the manuscript provide a valid rationale for the proposed study, with clearly identified and justified research questions?

Reviewer #1: Yes

2. Is the protocol technically sound and planned in a manner that will lead to a meaningful outcome and allow testing the stated hypotheses?

Reviewer #1: Yes

3. Is the methodology feasible and described in sufficient detail to allow the work to be replicable?

Reviewer #1: Yes

4. Have the authors described where all data underlying the findings will be made available when the study is complete?

The PLOS Data policy requires authors to make all data underlying the findings described in their manuscript fully available without restriction, with rare exception, at the time of publication. The data should be provided as part of the manuscript or its supporting information, or deposited to a public repository. For example, in addition to summary statistics, the data points behind means, medians and variance measures should be available. If there are restrictions on publicly sharing data—e.g. participant privacy or use of data from a third party—those must be specified.requires authors to make all data underlying the findings described in their manuscript fully available without restriction, with rare exception, at the time of publication. The data should be provided as part of the manuscript or its supporting information, or deposited to a public repository. For example, in addition to summary statistics, the data points behind means, medians and variance measures should be available. If there are restrictions on publicly sharing data—e.g. participant privacy or use of data from a third party—those must be specified.

Reviewer #1: Yes

5. Is the manuscript presented in an intelligible fashion and written in standard English?

Reviewer #1: Yes

You may also provide optional suggestions and comments to authors that they might find helpful in planning their study.

Reviewer #1: The manuscript is methodologically rigorous and fills a critical gap in the intersection of social work and ICTs research, holding significant theoretical and practical value. To further enhance its academic impact, improvements could be made in the following areas:

1、Methodology

(1)Search Strategy

The manuscript adheres to the Arksey & O’Malley framework and PRISMA-P guidelines, demonstrating a robust methodological foundation. But, also need Include concrete examples of search strings (e.g., for Sociological Abstracts) and explain optimizations (e.g., synonym expansion).

(2)Geographical Restrictions

The rationale for limiting empirical studies to the U.S. and Canada requires stronger justification. Clarify how these regions’ community practice theories and practices align theoretically to avoid exclusionary biases.

(3)Temporal Scope:

The 2015–2025 timeframe risks omitting pre-pandemic or foundational studies. Address how this choice might influence the comprehensiveness of findings.

2、Data analysis

(1)Quantitative Analysis

Specify hypothesis testing methods (e.g., chi-square tests, t-tests) for bivariate analyses to strengthen rigor.

(2)Qualitative Analysis

Elaborate on thematic analysis procedures, including coding processes, theme generation methods, and intercoder reliability checks.

.

Reviewer #1: No

---

## [Author Response · Author response to Decision Letter 1]

23 Mar 2026

Dear Editor-in-Chief Wei Lang and the PLOS One Editorial Board,

On behalf of my co-authors, PhD Candidate Ali Pearson and Professor Stephanie Begun, I am submitting the enclosed revised Study Protocol for review by the PLOS One editorial Board. We thank you for your thorough revisions and are confident that your comments will improve the final manuscript.

Reviewer #1:

The manuscript is methodologically rigorous and fills a critical gap in the intersection of social work and ICTs research, holding significant theoretical and practical value. To further enhance its academic impact, improvements could be made in the following areas:

Methodology

(1) Search Strategy: The manuscript adheres to the Arksey & O’Malley framework and PRISMA-P guidelines, demonstrating a robust methodological foundation. But also needs to include concrete examples of search strings (e.g., for Sociological Abstracts) and explain optimizations (e.g., synonym expansion).

Response: Thank you for this comment. We elaborated our search strategy optimization methods (e.g., thesaurus and keyword search, truncations, wildcards) under the subtitle “Identifying Relevant Literature” in the Methods section. An example of a complete search string is attached, as supporting information (S1). The string is too long to be included in the manuscript itself.

(2) Geographical Restrictions: The rationale for limiting empirical studies to the U.S. and Canada requires stronger justification. Clarify how these regions’ community practice theories and practices align theoretically to avoid exclusionary biases.

Response: Thank you for this comment. We provided a stronger justification for limiting empirical to the U.S. and Canada under “Screening Eligible Studies” in the Methods section. We also address the potential limitations posed by these geographical restrictions in our Discussion section.

(3) Temporal Scope:

The 2015–2025 timeframe risks omitting pre-pandemic or foundational studies. Address how this choice might influence the comprehensiveness of findings.

Response: We mentioned the pandemic as a reference point to justify the 2015–2025 timeframe to demonstrate that this timeframe can also be productive in tracing the impact of COVID-19 on the relationships between ICTs and community practice. While findings may suggest insights into the impact of the pandemic on the relationship between community practice and ICTs, this is not the primary question asked in this review. Accordingly, we provided a more detailed rationale for this timeframe, anchoring it in relation to the Social Work Grand Challenges framework (specifically Harness Technology for Social Good).

Data analysis

(1) Quantitative Analysis:

Specify hypothesis testing methods (e.g., chi-square tests, t-tests) for bivariate analyses to strengthen rigor.

Response: We elaborated on hypothesis testing methods in Data Analysis and Synthesis in the Methods section.

(2) Qualitative Analysis:

Elaborate on thematic analysis procedures, including coding processes, theme generation methods, and intercoder reliability checks.

Response: We elaborated on thematic analysis procedures in Data Analysis and Synthesis in the Methods section.

Additional Changes #:

(1) Refining Conceptualization: Based on the community social work literature, we replaced the expression “grassroots organizing/advocacy” with “community organizing”. In the North American social work literature, the term “community organizing” is widely used to encompass local grassroots organizing, organizational coalition building, and participatory advocacy.

(2) Proofreading & Clarifications: We thoroughly proofread and added clarification in several places throughout the manuscript. All changes can be traced in the track-changes version of the manuscript.

Sincerely,

Ofir Sivan, MSW, RSW

PhD Student, Factor-Inwentash Faculty of Social Work, University of Toronto, Ontario, Canada

---

## [Decision Letter · Decision Letter 1]

25 Mar 2026

Protocol for a Scoping Review on Information and Communication Technologies (ICTs) in Community Practice

PONE-D-25-55788R1

Dear Dr. Sivan,

We’re pleased to inform you that your manuscript has been judged scientifically suitable for publication and will be formally accepted for publication once it meets all outstanding technical requirements.

Kind regards,

Wei Lang, Ph.D.

Academic Editor

PLOS One

Additional Editor Comments (optional):

Reviewers' comments:

Reviewer's Responses to Questions

**Comments to the Author**

1. Does the manuscript provide a valid rationale for the proposed study, with clearly identified and justified research questions?

Reviewer #1: Yes

2. Is the protocol technically sound and planned in a manner that will lead to a meaningful outcome and allow testing the stated hypotheses?

Reviewer #1: Yes

3. Is the methodology feasible and described in sufficient detail to allow the work to be replicable?

Reviewer #1: Yes

4. Have the authors described where all data underlying the findings will be made available when the study is complete?

The PLOS Data policy requires authors to make all data underlying the findings described in their manuscript fully available without restriction, with rare exception, at the time of publication. The data should be provided as part of the manuscript or its supporting information, or deposited to a public repository. For example, in addition to summary statistics, the data points behind means, medians and variance measures should be available. If there are restrictions on publicly sharing data—e.g. participant privacy or use of data from a third party—those must be specified.requires authors to make all data underlying the findings described in their manuscript fully available without restriction, with rare exception, at the time of publication. The data should be provided as part of the manuscript or its supporting information, or deposited to a public repository. For example, in addition to summary statistics, the data points behind means, medians and variance measures should be available. If there are restrictions on publicly sharing data—e.g. participant privacy or use of data from a third party—those must be specified.

Reviewer #1: Yes

5. Is the manuscript presented in an intelligible fashion and written in standard English?

Reviewer #1: Yes

You may also provide optional suggestions and comments to authors that they might find helpful in planning their study.

Reviewer #1: The revisions not only addressed the key concerns raised by the reviewers but also strengthened the scientific rigor and methodological soundness of the study.

.

Reviewer #1: No

---

## [Editor Report · Acceptance letter]

PONE-D-25-55788R1

PLOS One

Dear Dr. Sivan,

I'm pleased to inform you that your manuscript has been deemed suitable for publication in PLOS One. Congratulations! Your manuscript is now being handed over to our production team.

Kind regards,

on behalf of

Dr. Wei Lang

Academic Editor

PLOS One